# Fractal analysis of urban catchments and their representation in semi-distributed models: imperviousness and sewer system

Auguste Gires[1], Ioulia Tchiguirinskaia[1], Daniel Schertzer[1], Susana Ochoa Rodriguez[2], Patrick Willems[3], Abdellah Ichiba[1,4], Li-Pen Wang[3], Rui Pina[2], Johan Van Assel[5], Guendalina Bruni[6], Damian Murla Tuyls[3], Marie-Claire ten Veldhuis[6]

[1]HMCo, École des Ponts, UPE, Champs-sur-Marne, France
[2]Urban Water Research Group, Department of Civil and Environmental Engineering, Imperial College London, Skempton Building, London SW7 2AZ, UK
[3]Hydraulics Laboratory, KU Leuven, 3001, Heverlee (Leuven), Belgium
[4]Conseil Départemental du Val-de-Marne, Direction des Services de l'Environnement et de l'Assainissement (DSEA), Bonneuil-sur-Marne, 94381, France
[5]Aquafin NV, Dijkstraat 8, 2630 Aartselaar, Belgium
[6]Department of Water Management, Faculty of Civil Engineering and Geosciences, Delft University of Technology, PO Box 5048, 2600 GA Delft, the Netherlands

*Correspondence to*: Auguste Gires (auguste.gires@enpc.fr)

**Abstract.** Fractal analysis relies on scale invariance and the concept of fractal dimension enables to characterise and quantify the space filled by a geometrical set exhibiting complex and tortuous patterns. Fractal tools have been widely used in hydrology but seldom in the specific context of urban hydrology. In this paper fractal tools are used to analyse surface and sewer data from 10 urban or peri-urban catchments located in 5 European countries. The aim was to characterise urban catchment properties accounting for the complexity and inhomogeneity typical of urban water systems. Sewer system density and imperviousness (roads or buildings), represented in rasterized maps of 2 m x 2 m pixels, were analysed to quantify their fractal dimension, characteristic of scaling invariance. The results showed that both sewer density and imperviousness exhibit scale invariant features and can be characterized with the help of fractal dimensions ranging from 1.6 to 2, depending on the catchment. In a given area consistent results were found for the two geometrical features, yielding a robust and innovative way of quantifying the level of urbanization. The representation of imperviousness in operational semi-distributed hydrological models for these catchments was also investigated by computing fractal dimensions of the geometrical sets made up of the sub-catchments with coefficients of imperviousness greater than a range of thresholds. It enabled to quantify how well spatial structures of imperviousness were represented in the urban hydrological models.

## 1 Introduction

The aim of this paper is to consistently characterise urban catchment properties accounting for the complexity and inhomogeneity typical of urban water systems. It is focused on two main properties of urban catchments, namely the

geometry of the sewer system and the distribution of impervious surfaces. Such characterisation is important to obtain insights in the urban catchment response behaviour at the various spatial scales that control the relation between rainfall and sewer flows; to develop convenient methods that allow evaluation of the urban catchment characteristics implemented in urban drainage models (the ones that are of importance for obtaining reliable spatially variable urban catchment responses;

e.g. spatial imperviousness structure); to develop method that support the urban hydrological modeller in the decision about the spatial details required to obtain reliable model (impact) results. Achieving this has proved to be difficult using traditional tools, mostly based upon Euclidean geometry, due to the variability and inhomogeneity in catchment characteristics (ex among other Berne et al, 2004). An alternative to traditional tools could be the use of fractal geometry (Mandelbrot, 1983), which relies on the concept of scale invariance, i.e. similar structures are visible at all scales. The

concept of fractal dimension enables to characterize in a scale invariant way the space filled by a geometrical set in its embedding space. Fractal analysis and more generally scaling analysis have been often and successfully used in geophysics, including hydrology, but seldom in the specific context of urban hydrology.

For example, they have been used to characterise river networks, including quantification of main stream sinuosity (Nikora , 1991; Hjeimfeit, 1988), quantification of how the network fills space (La Barbera and Rosso, 1989; Takayasu, 1990;

Foufoula-Georgiou and Sapozhnikov, 2001; Gangodagamage et al., 2011, 2014,), and simultaneous quantification of both features (Tarboton et al., 1988; Rosso et al., 1991; Tarboton, 1996; Veltri et al. 1996). River basins have also been analysed with fractal analysis. For instance, Bendjoudi and Hubert (2002) showed that the perimeters of the Danube (Eastern Europe) and Seine (France) river basins are too tortuous to be scale-independent. Rainfall occurrence patterns also appear to exhibit fractal features (Lovejoy and Mandelbrot, 1985; Olsson et al., 1993; Hubert et al., 1995). In extensions including the use of

multifractal tools, i.e. for fields and not simply geometrical shapes, such tools have also been used to study river discharges and rainfall time series (see Tessier et al., 1996, or Pandey et al., 1998, for examples combining both).

Some authors relied on the same concept of fractal dimension for characterizing land use cover in various contexts. For example Cheng et al. (2001) computed a fractal dimension for various land use classes and used it to analyse land use change between two areal pictures taken 20 years apart over a 4 km$^2$ mountainous catchment. Darrel and Wu (2001) computed

fractal dimensions of three land use classes -desert, agriculture and urban- and used it to analyse their evolution during a century over a 69 km x 89 km area around Phoenix (Arizona, United States of America). This allowed investigating the effect of urbanization over landscape and was used to develop a model to reproduce observed features. Similarly, Tannier et al. (2011) used this concept to identify the morphological boundary of urban areas in a scale invariant way. Iverson (1988) estimated fractal dimensions for numerous land use types to study the evolution of landscape over 160 years in Illinois

(United States of America). Soil features have also been studied with fractal analysis. For instance Wang et al. (2006) analysed particle size distribution with fractal concepts. A feature emphasized by many authors is the relationship between fractal features and power law decay (i.e. non Gaussian behaviour) of various fields such as river portion length, rainfall event duration, particle size distribution or distance between buildings (Mandelbrot, 1983; Lavergnat and Golé, 1998; Tarboton, 1996; Wang et al., 2006; Tannier et al., 2011). This implies that up- and downscaling of meteorological and

hydrological parameters needs to account for this non-Gaussian behaviour. For hydrological analysis it means that hydrological models are likely to be sensitive to scale differences between rainfall input and catchment characterisation (Ogden and Julien, 1994).

Despite this wide range of applications, fractal analysis has seldom been used to specifically address the topic of urban hydrology. Initial attempts to characterize urban drainage networks (Sarkis, 2008; Gires et al., 2014) or imperviousness (Gires et al. 2014) have been carried out on limited areas. In this paper we go a step further and implement fractal analysis on 10 urban catchments with different characteristics located across 5 European countries. The investigation includes analysis of the sewer network geometry and distribution of imperviousness derived from available GIS data, including the way in which it is represented in operational semi-distributed hydrodynamic urban drainage models. In order to be able to use the same technique to analyse both sewer networks and map of distributed imperviousness, we use fractal tools on them, and not multifractal ones as the one found in (De Bartolo et al. 2004, 2006) for river networks. Multifractals will be used in the characterization of the representation of imperviousness in models. This multi-catchment investigation allows obtaining robust results which are representative of a range of hydrological characteristics. The opportunity to carry out this multi-catchment investigation arose from the Interreg North West Europe (NWE) project RainGain, which focuses on improving rainfall estimation and pluvial flood modelling and management in urban areas across NWE.

The paper is organised as follows. In section 2 the available dataset over the 10 pilot catchments is described. The concept of fractal dimension and the methodology used to compute it are explained in section 3. Results are presented and discussed in section 4. In section 5 the main conclusions are presented and future work is discussed.

## 2 Experimental sites and datasets

Ten urban catchments, with areas in the range of $2 - 8$ km$^2$ and located in five European countries (UK, France, the Netherlands, Belgium and Portugal) were adopted as pilot sites in this study. The general location of the pilot catchments is shown in Figure 1 and their main characteristics are summarised in Table 1.

For each pilot catchment three types of data are analysed in this paper and Fig. 2 displays them for all the catchments :

(i) The sewer system, which is considered as a network of linear pipes (left column in Fig. 2). The level of precision of available data is not the same for all the catchments. Indeed for the Morée-Sausset and Torquay catchments, only the main pipes are taken into account, whereas for the other all pipes down to street level (not the connections from building or houses to the network) are available

(ii) An imperviousness map at a resolution of 2 m x 2 m generated with the help of QGIS (www.qgis.org) based on data derived mainly from Open Street Map (http://www.openstreetmap.org/) (middle column in Fig. 2). More precisely, for each catchment the road layer (of polyline type) was retrieved from the Open Street Map platform and a 4 m buffer (adopted based on normal width of roads in urban and peri-urban catchments) was set on both sides of this polyline layer. The

building layer was retrieved either from the same platform or from local building register datasets. These two data sets were rasterized in a map with pixels of size 2 m x 2 m. An imperviousness map was then derived in which a pixel containing roads or buildings is marked as impervious and other pixels are marked as pervious.

(iii) A map of imperviousness derived from catchment representation in semi-distributed hydrodynamic models (right column in Fig. 2). A validated operational semi-distributed hydrodynamic model was available for each of the pilot catchments, except for Jouy-en-Josas. In this type of model the whole catchment is split into a number of sub-catchment, an independent hydrological block corresponding to a portion of the full catchment. The models are not the same for all the pilot sites but they all function with the same underlying principles. Each sub-catchment contains a mix of pervious and impervious surfaces whose runoff drains to a common outlet point, which could be either a node of the drainage network or another sub-catchment (Rossman, 2010). Each sub-catchment is characterised by a number of parameters, including total area, length, slope, proportion of each land use and soil type characteristics. Rainfall is inputted as homogeneous in space within each sub-catchment, and based on the sub-catchment's characteristics, the total runoff is estimated with the help of a lumped model and routed to the outlet point. The flow in pipes is then represented with the help if numerical approximation of 1D shallow water equations. The size and distribution of sub-catchments depend on the modeller's choices according to the local features, the available data and desired level of precision, Based on the percentage of impervious areas assigned to each sub-catchment within each pilot catchment, a raster map with pixels of size 2 m x 2 m was generated for each pilot site. The distribution of sub-catchments is visible in Fig. 2 because the values of imperviousness are uniform over them. Average size of sub-catchment elements varies greatly according to the studied area (see Table 1). For instance, it is much greater in Sucy-en-Brie than in Rotterdam-Kralingen. The purpose of the paper is not to evaluate the performance of those models all previously validated and used operationally by practitioners but to characterize their inputs, notably in comparison with more refined impervious data maps. Discussions on outputs of these models can be found in Ochoa-Ridriguez et al. 2015.

**3 Methodology**

As explained in section 1, the concept of fractal dimension was used in this paper to characterize various geometrical sets ( namely the sewer network and imperviousness), embedded in a 2-dimensional space. Let's consider such a bounded set *A* of outer scale $l_0$. The first step consists in changing its resolution, i.e. modifying its observation scale *l*. The resolution $\lambda$ is defined as the ratio between the outer scale and the observation scale ($\lambda = \dfrac{l_0}{l}$). This is achieved by representing it with the help of non-overlapping pixels of size *l*. At a given scale the set *A* is represented by all pixels overlaying the geometrical set. A range of values is tested for *l*. In this study, the analysis started at the smallest pixel size available, i.e. 2 m. The pixel size is then multiplied by two at each step, i.e. four adjacent pixels are merged, up to a maximum pixel size which covers as much of the total catchment area as possible. An illustration of this process for the sewer system of the Herent case is displayed in Fig. 3. Limited differences are visible when changing the observation scale from 2 m to 4 m (some details are

lost in the intersections, and close pipes merged), and they are much more pronounced with observation scales equal to 16 m and 64 m (merging of numerous pipes). These observations are actually consistent with the scale break at 64 m that will be identified and discussed in section 4.1.

This means that the outer scale of the studied set will necessarily be the original pixel size multiplied by a power of two, closest to the maximum catchment scale (pixels are merged 4 by 4 in order to maximise the number of points in the following linear regression; less reliable results would be obtained with by merging pixels 9 by 9 or 25 by 25). As a consequence, square areas are extracted from the studied catchments to be analysed with the help of fractal analysis. Their size is chosen as a balance between achieving the greatest possible coverage (which increases the range of available scales) and limiting the portion of the square extending outside the catchment boundary (given that the artificial zeros in these portions might bias the analysis due to side effects). The studied areas within each catchment are shown in Fig. 2 for all catchments. In four catchments (Cranbrook, Ghent, Herent and Torquay) two areas are studied, sometimes slightly overlapping (Cranbrook and Ghent).

Now that the methodology to change the resolution of the data set was explained, it is possible to describe the computation of its fractal dimension with the help of the box-counting method (Hentschel and Proccacia, 1983; Lovejoy et al. 1987). Let $N_{\lambda,A}$ be the number of non-overlapping pixels of size $l$ necessary to cover the set $A$. For a fractal object this number and the resolution are power-law related in the high resolution limit ($\lambda \to +\infty$), with an exponent equal to the fractal dimension ($D_F$) of the set; i.e. we have:

$$N_{\lambda,A} \approx \lambda^{D_F} \quad (1)$$

A standard technique to estimate a fractal dimension is the box-counting one which relies on the previous equation. To implement this technique, one defines non-overlapping pixels of size $l$ as explained in the previous paragraph and plots Eq. 1 on a log-log scale. For a fractal set the points will be along a straight line which slope is equal to $D_F$. The quality of the scaling is assessed with the help of the coefficient of determination $r^2$ of the linear regression. It is an imperfect indicator, especially given the limited number of points available, and should be completed by visual inspection, The fractal dimension quantifies the sparseness of the set $A$, i.e. how much space it fills across scales.

The notion of fractal dimension is well suited for studying binary field such a sewer network or map of imperviousness. However when the field can have more than two states, as it is the case in this paper for the maps of representation of imperviousness inputted in semi-distributed hydrodynamics models, multifractals tools might be needed. Intuitively such fields are characterized with the help of various fractal dimensions, i.e. for each threshold, the geometrical set of the areas where the field exceeds it exhibits a different fractal dimension. More rigorously the notion of threshold, which is scale dependent, is replaced by the scale invariant one of singularity, $\gamma$. Then and the portions of a multifractal field $\varepsilon_\lambda$ where it exceeds the threshold $\lambda^\gamma$ at a given resolution $\lambda$ are studied. Their probability scales as:

$$\Pr\left(\varepsilon_\lambda > \lambda^\gamma\right) \approx \lambda^{-c(\gamma)} \quad (2)$$

Where $c(\gamma)$ is the co-dimension function which fully characterizes the variability not only at a single scale but across scales of $\varepsilon$ (see Schertzer and Lovejoy 2011 and references therein for a recent review of this formalism). $c(\gamma)$ corresponds to the fractal co-dimension (equal to the embedding Euclidian dimension – 2 here – minus the fractal dimension) of the geometrical set where $\varepsilon_\lambda$ exceeds $\lambda^\gamma$. In the specific framework of Universal Multifractals (Schertzer and Lovejoy 1987, 1997), the co-dimension function only depends on three parameters which have a physical interpretation: $H$ the non conservation parameter which measures the scaling behaviour of the mean of the studied field ($\langle \varepsilon_\lambda \rangle \approx \lambda^H$, $H$=0 for a conservative field), $C_1$ the mean intermittency which measures the clustering of the average intensity (mathematically it is $c(\gamma_1)$ where $\gamma_1$ is the singularity corresponding to the mean; $C_1$=0 for an homogenous field); and $\alpha$ the multifractality which measures how the mean intermittency evolves when considering singularities slightly different from $\gamma_1$ ($\alpha$=0 for a fractal field). These parameters are estimated with the help of the Double Trace Moment Technique (DTM) (Lavallée et al, 1993).

## 4 Results and discussion

### 4.1 Sewer network and distributed land use

Figure 4 shows a log-log plot of $N(\lambda)$ versus $\lambda$ (Eq. 1) for the Torquay North case study. A single scaling behaviour over the whole range of available scales is not retrieved. Indeed, the plot exhibits a scale break at roughly 64 m pixel scale, separating two distinct scaling regimes. Over each regime, the scaling is robust with $r^2$ all above 0.99, and visible straight lines. Similar qualitative features, i.e. two distinct well defined scaling regimes separated by a break, are retrieved for the other studied areas and not displayed. Numerical values of the computed fractal dimensions and the values of scale break for all studied area are reported in Table 2.

For the scaling regime associated with small scales (i.e. right portion of the graph), a fractal dimension basically equal to 1 is found for all the study areas. This does not contain information on the network's features but simply reflects the linear structure of the pipes at these scales. It also means that the maximum resolution of the available data (2 m pixels here) is not critical to the analysis and does not introduce a potential bias. Indeed increasing or decreasing it would simply yield to extending or shrinking the widths the scale range of this regime but will not affect the values at larger scales discussed below. The break is located at roughly 64 m for most of the areas, which is consistent with the distance between two streets. It is at 32 m in Coimbra and Rotterdam-Centrum which correspond to densely urbanized city centres. The break at 128 m for the Moree-Sausset sewer is due to the fact that only major sewer pipes are available and included in the numerical network model meaning small scale features simply extend over wider range of scales. Including more pipes would likely lead to shifting the scale break to smaller scales. It appears that for all the catchments the break is observed at roughly the approximate inter-pipe distance of the portion of network taken into account. For the large scales regime (~ 64 m to 2048 m),

an actual fractal dimension between 1 and 2 characterizing the space filled by the network is retrieved. According to catchment we find $D_F$ ranging from 1.69 to 1.94. With smaller scales, this regime is expected to continue until the physical scales of structures is reached below which a fractal dimension of 2 would obviously be found. It is in any case smaller than two meaning that the network does not completely fill the 2D-space. An interpretation of these values is that are representative of the level of urbanization of the areas. For instance, we find the greater fractal dimensions in the Rotterdam districts and smaller ones in less-urbanised Jouy-en-Josas and Torquay. This will need to be confirmed with the analysis of imperviousness maps.

These results are consistent with values found in similar studies for drainage networks. Sarkis (2008) found a fractal dimension equal to 1.67 for the pluvial drainage network of the Val-de-Marne County (South-East of Paris), based on an analysis at scales of 290 m to 18 km, only considering the main pipe network. Typical values for natural river network fractal dimensions (computed with the box counting technique) are usually smaller than those found here for urban catchments. For instance Takayasu (1990) found $D_F$ for the Amazon and Nile Rivers equal to 1.85 and 1.4 respectively.

Figure 5 displays the impervious pixels (in blue), along with the computation of the fractal dimension of the corresponding geometrical set for the Torquay North area. It appears that a unique scaling regime on the whole range of available scales is identified (single straight line), resulting in fractal dimension 1.81. Unique scale regimes are also found for impervious surface distributions in the all other studied areas. The scaling regime is robust with visible straight lines as in Fig. 5 (right) and $r^2$ always greater than 0.995. The uniqueness of the regime also means that results are not sensitive to the initial pixel size of 2 m as for the sewer system analysis (but for a different reason). Increasing this size would simply reduce the width of the range of scales available to compute the fractal dimension but not change its value. Numerical values of these fractal dimensions are reported in Table 2. Despite the fact that the impervious pixels do not represent at a 2 m resolution the majority of the pixels, their fractal dimension is rather elevated meaning that the impervious areas fills the space in urban areas. As expected less urbanised areas exhibit lower fractal dimension.

For a given catchment, numerical values of fractal dimension for distributed imperviousness are similar to the ones found at large scales in the sewer system analysis. Discrepancies are usually smaller than 0.1; smaller than the differences between the various catchments. Areas of similar urban density have similar fractal dimensions and lower density urban areas are consistently characterised by lower fractal dimensions. These numerical similarities are worth noting and actually one of the main finding of this analysis, confirmed on a wide set of study areas. Indeed it suggests that the scaling behaviours observed on sewer networks and distributed land use have the same physical basis and reflect a unique underlying level of urbanisation. The only difference being that it stops at the inter-pipe distance for the sewer network whereas it expands down to 2 m scale for the imperviousness. Contrary to other formalisms such as the use of a single percentage of imperviousness defined with data at an arbitrary scale, this fractal dimension is a quantity valid across scales and furthermore based on the characterization of two aspects related to urbanization (namely the sewer network and the distributed imperviousness) which makes it robust.

## 4.2 Representation of imperviousness in semi-distributed models

After having investigated the fractal behaviour of sewer system and imperviousness with the help of distributed data, the imperviousness distribution used in operational semi-distributed hydrodynamic models is studied in this section. A given threshold $T$ is selected and fractal features of the geometrical sub-set made up of the sub-catchments with imperviousness greater than the threshold $T$, representing different degrees of imperviousness in this case, are analysed. Figure 6 illustrates the corresponding sub-sets and computation of the fractal dimensions for $T$ equal to 20, 50 and 80% for the Torquay North study area. Figure 7 displays $r^2$ (coefficient of determination of the linear regressions defining $D_F$) vs. $T$ (top) and $D_F$ vs. $T$ (bottom) for all pilot areas.

As expected, at higher thresholds, the remaining impervious areas are smaller and the associated fractal dimensions are also smaller. It should be noted that the quality of the scaling also tends to diminish for increasing imperviousness thresholds. This effect is significant for some areas such as Moree-Sausset, Herent and Sucy-en-Brie and hence limits the possible interpretation of this analysis. In these cases, there is a very limited (one or sometimes even zero) number of remaining sub-catchments at high imperviousness thresholds, which is likely to bias the analysis. This phenomenon is due the smaller number of sub-catchment in these cases. The most critical one is that of Sucy-en-Brie, for which the model consists of only eight sub-catchments (see Fig. 2). Such low spatial resolution hampers implementation of fractal analysis and this is reflected in the low $r^2$ for thresholds greater than 40% (no data for T>60%). Computations on larger areas, that would include more sub-catchments or a higher model resolution (smaller sub-catchment size and greater number of sub-catchments as done in other study areas) with high degree of imperviousness (as it is the case for the Rotterdam-Centrum study area), would be needed to confirm this interpretation. This issue illustrates the need for models with a number of sub-catchment enabling to fully represent the variability of imperviousness. The use of fully distributed models is a way to improve this representation. For hydrological purposes the use of more distributed model also enable to better account for the spatio-temporal rainfall variability which is known to have a significant impact on simulated outputs (Gires et al. 2014).

Interestingly, the fractal dimension estimates are in overall agreement with the level of urbanization discussed in the previous section, i.e. the most urbanized areas exhibit the greatest fractal dimension for all thresholds. This is especially true for thresholds lower than 60%. For greater ones, whose estimates are less reliable, more differences are noted. For instance $D_F$ with T>60% for London-Cranbrook are much smaller than for Ghent whereas the estimates from the distributed data are rather close (Table 2). This reflects different choices by the modellers in the representation of the urban catchment. Indeed, imperviousness is one of the main 'tuning' variables used in the calibration of urban drainage models. The differences in imperviousness observed between semi-distributed models and distributed datasets may be caused by 'lumping' of catchment characteristics in the models and errors in the model and/or in the distributed datasets. This effect also partially explains the fact that disparities between the catchments tend to strengthen with increasing thresholds which are likely to be more affected by modellers' choices. Another possible explanation that would need to be further confirmed by analysis on a larger number of data sets is simply that the spatial structure of the highly impervious areas could exhibit a clear difference with

regards to less urbanised ones (see also multifractal analysis). It should be mentioned that similar to the findings of the previous section, estimates obtained for various areas within a given catchment are rather similar, except for Herent. In Herent the impervious areas fill a greater space in the East study area than in the West one, which was not the case for the imperviousness from the distributed data. This is explained by different modelling choices with respect to the level of detail in catchment representation. Models could also have been calibrated long time before the GIS data was obtained. For Coimbra the differences, especially for low thresholds, are smaller than the ones observed on the sewer system and the distributed imperviousness.

Given that we found that the fractal dimension of sub-catchments' imperviousness of the semi-distributed models was dependent on the threshold used to define it, we naturally investigated the possibility of using a multifractal framework to analyse this dependency. This is achieved by checking the adequacy of the empirical co-dimension function $c(\gamma)$ with its theoretical expected shape. More precisely, at the maximum resolution $\Lambda$, for each studied threshold $T$, the corresponding singularity $\gamma_T$ is estimated as $\log_\Lambda \dfrac{T}{\langle T \rangle}$ ,where $\langle T \rangle$ is the average of the studied thresholds and equal to 50 here. The empirical value of $c(\gamma_T)$ is then simply given by the fractal co-dimension ($2$-$D_F$). Finally $2$-$D_F$ is plotted as a function of $\gamma_T$, along with the theoretical shape of $c(\gamma)$. This technique is known as functional box-counting in the literature (Lovejoy et al. 1987). The UM parameters $\alpha$ and $C_1$ used are those retrieved from DTM analysis and reported in Table 2. They are generally in the range 1.2-1.6 for $\alpha$ and 0.01-0.09 for $C_1$. The quality of the scaling related to $\alpha$ and $C_1$ is low with coefficient of determination in the linear regressions of the order 0.8-0.9, meaning that their reliability is not very high. Figure 8 displays these curves for four representative cases. It should be mentioned that the theoretical curve of $c(\gamma)$ was shifted horizontally "manually" to better fit the empirical points. This mimics the effect of $H$, which was not possible to estimate robustly with this data set. It appears that the agreement between the empirical points and theoretical expectations is good in most of the cases (Herent West, Cranbrook and Torquay on Fig. 8), and it remains valid on a large range of $c(\gamma)$. In other cases such as Coimbra West, it is less good and some discrepancies are visible. These results should only be taken as preliminary ones that should be confirmed by further analysis on extended data sets given the limitations of this study: small range of available scales, low quality of the data which is not actual physical data but a representation with different resolution in models, and manual fitting of $H$. In some cases such as Torquay North and to a smaller extent Herent West in Fig. 8, there seems to be a linear behaviour for empirical points associated with large singularities. This is the signature of a multifractal phase transition which reflects the large scale influence of small scale variability. Such behaviour is commonly found in geophysical fields. It is associated with a power-law tail for the probability distribution of the pixels' imperviousness. Results are not reliable enough to get definitive conclusions, but they are encouraging and should be a first step before a more in-depth analysis of the notion of imperviousness and its characterization in a scaling framework. A

possible useful application would be the possibility to easily and realistically fill gaps of missing data in imperviousness maps.

Finally, fractal dimensions of the imperviousness computed for the semi-distributed models were compared to those derived
from fully distributed GIS-data (section 4.1). This is done in Fig. 9 for three studied areas. $D_F$ vs. $T$ for the model is plotted (same as in Fig. 7 bottom) along with the fractal dimension from the distributed data (horizontal line) and the percentage of impervious pixels with 2 m size pixels (vertical line). If the spatial distribution of the average catchment imperviousness is realistically represented in the model, the intersection of these two straight lines should be located on the $D_F$ vs. $T$ curve. This is clearly visible in Fig. 8 for Morée-Sausset and Herent West; much less for Cranbrook. The location of the
intersection of the two straight lines below the curve indicates that the Cranbrook model overestimates space filled by the areas with imperviousness greater than the average. In order to quantify this effect, the difference (denoted % $_{diff}$) between the value of $T$ at the intersection of the $D_F$ vs. $T$ curve with the horizontal line and the percentage of impervious pixels is reported in Table 2. The absolute value of this difference is always smaller than 18% and smaller than 10% in 5 cases. There is no obvious relation between the numerical value of this quantity and the level of resolution of the hydrodynamic model.

The percentages of distributed imperviousness (%) at the highest resolution $\Lambda$ and of the imperviousness of semi-distributed models (%+%$_{diff}$) could be compared to the percentages of imperviousness resulting from the fractal dimension estimates: $\%_{D_F} = 100\Lambda^{D_F-2}$. Figure 10 displays the results of such a comparison. First of all, this figure (Fig. 10.a) demonstrates that for several catchments uncertainties in scaling estimates result in visible discrepancies between (%) and ($\%_{D_F}$) that are expected to be identical in the case of a "perfect" scaling. The difference of these two estimates is based on the fact that the
percentages of distributed imperviousness (%) is computed at the highest resolution $\Lambda$ only, whereas the fractal dimension estimates are computed across all the scales and hence result in a multiscale characteristic for each catchment. Then, the adjusted percentage of the imperviousness of semi-distributed models, in general, diverges even stronger w.r.t. the one resulting from the fractal dimension estimates. The only two improvements were observed for the Rotterdam-Kralingen and Herent West catchments (see Fig.10.b).

Such analysis could support validation of the representation of catchments in semi-distributed models; the smaller the difference, the better catchment imperviousness is represented by the model. It should be mentioned that this interpretation assumes that data available for analysing distributed imperviousness is accurate and complete, which is generally supported by the scaling behaviour of the data.

### 4.3 Representation of imperviousness of buildings

In this sub-section we discuss the results of the comparison of fractal dimensions computed on two different geometrical sets: the total imperviousness areas as roads and buildings ($D_{F\_all}$) and the buildings only ($D_{F\_build}$). Obtained results show that

for each catchment the geometrical set of buildings alone behaves as a fractal set. Indeed as for the analysis carried out in section 4.1 (total imperviousness) straight lines are found in the linear regression of Eq. 1 in log-log plot (not shown) with $r^2$ remaining greater than 0.99, meaning that numerical values of fractal dimensions are robust. Obviously $D_{F\_build}$ could not be greater than $D_{F\_all}$, since the building areas are embedded within a larger fractal set of all impervious areas, and we have

$N_{build} = \lambda^{D_{F\_build}} = \lambda^{aD_{F\_all}}$ . The empirical results displayed on Fig.11 suggest that a common value $a=0.945$ remains suitable for the majority of the catchments. Such small coefficient may influence the scaling at the smallest scales only. The changes seem to increase with smaller values either meaning that the network of road has a greater importance in these cases, or simply due to a slight decline of scaling. Indeed, by comparing Figures 10 and 11, one may note a slight amplification of scaling issues compare to those observed for the percentages of distributed imperviousness.

This analysis was made to investigate the relationships between the fractality of building distributions, as a source for potential green roof implementation for water flow management, within fractality of the whole imperviousness areas. Indeed green roof are one of the available tools that can be used to optimize (if needed) water flows in urban and peri-urban areas, hence a need to understand better their potential distribution. More precisely, to increase the functionality of green roofs over the full range of catchment scales (Versini et al., 2016), an optimization of green roof locations could be maid to increase

their fractal dimension up to the fractal dimension of the total imperviousness area. The fractal tools could be also used to evaluate the potential impact of green roofs.

**5 Conclusions**

In this paper we implemented (multi-)fractal analysis in the context of urban hydrology on ten catchments located in five European countries. The results have consequences both in terms of urban catchment characterization and representation in

urban hydrological models.

First, it appears that the fractal dimension of either the sewer network or the impervious pixels (roads or houses) on a 2 m pixels map can be used to characterize the level of urbanization of a given area. In fact, for a given area similar estimates are obtained for both geometrical sets. The main difference is that the scale invariance is valid from one or few kilometres down to only approximately inter-pipe distance for the sewer network whereas it extends down to 2 m for imperviousness, which

matches with the spatial resolution of the imperviousness datasets. This tool is innovative in the context of urban hydrology, because it provides a quantitative estimate of a level of urbanization which is valid across scales and not only at the scale at which it is defined as for other tools. These findings open new practical perspectives that should be explored in future work. An example is the possibility to indentify consistent – across scales – areas that should be modelled separately. Another one is the possibility of relying on the scale invariance features to fill gaps of missing data in a realistic way. This issue is

increasingly visible as one goes toward higher resolution model. It is furthermore an acknowledgment of the complexity of the notion of imperviousness which is usually simplified in state-of-the-art urban hydrological models in which it is often

represented as a mere percentage, thus neglecting without taking into account its heterogeneous distribution. Using scale invariant concepts able to handle more appropriately these features is a lead that should used to innovatively improve distributed hydrological models.

Second, the representation of imperviousness in operational semi-distributed models was analysed. It appears that, by analysing the geometrical set made of sub-catchments with imperviousness greater than a given threshold, it is possible to retrieve urbanization patterns. In this study, it was found that fractal dimension values decrease from 1.9-2.0 for imperviousness degrees above 10% down to 1.4-1.6 for imperviousness degrees above 90%. Results for higher imperviousness degrees were subject to larger uncertainty as a result of data scarcity; findings should be verified in studies based on larger datasets.

It was also shown that comparing fractal dimension values related to modelled imperviousness to imperviousness represented in high resolution GIS datasets allows to quantify how well imperviousness is represented in urban hydrological models. These results open perspective for the development of tools to verify whether a hydrological model properly represents the degree of imperviousness in a catchment and also to study urbanisation patterns emerging at different degrees of imperviousness. Such insights could latter be used in support of hydrological analysis as well as other urban development analyses.

Acknowledgements

The authors greatly acknowledge partial financial support from European Union INTERREG IV NWE RainGain Project (www.raingain.eu). Authors acknowledge Julien Richard for his help in preparing the data and Figure 1.

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

Tables:

| | Catchment characteristics | | | | | Model characteristics | | |
|---|---|---|---|---|---|---|---|---|
| | Area [ha] | Length[a] [km] | Slope[b] [m/m] | Land use[c] | Pop. density [per/ha] | Total pipe length [km] | Num. of SC[d] | Mean / STD SC size [ha] |
| Cranbrook, UK | 865 | 6.10 | 0.0093 | R&C | 48 | 98 | 1765 | 0.49/0.71 |
| Torquay (Town Centre), UK | 570 | 5.35 | 0.0262 | R&C | 60 | 41 | 492 | 1.16/1.09 |
| Morée-Sausset, FR | 560 | 5.28 | 0.0029 | R&C | 70 | 15 | 47 | 11.92/10.34 |
| Sucy-en-Brie, FR | 269 | 4.02 | 0.0062 | R&C | 95 | 4 | 9 | 29.89/27.47 |
| Herent, BE | 511 | 8.16 | 0.0083 | R | 20 | 67 | 683 | 0.71/1.27 |
| Jouy-en-Josas, FR | 302 | 2.47 | 0.037 | R | 15 | - | - | - |
| Ghent, BE | 649 | 4.74 | 0.0001 | R | 24 | 83 | 1424 | 0.46/0.89 |
| Rotterdam - Kralingen, NL | 670 | ~ 2[e] | 0.0003 | R&C | 154 | 143 | 2435 | 0.12/0.13 |
| Rotterdam Centrum, NL | 340 | ~ 1[e] | 0.0001 | R&C | 88 | 140 | 2832 | 0.0769/ 0.0737 |
| Coimbra, PT | 158 | 4.21 | 0.0333 | R&C | 116 | 34.75 | 911 | 0.17/0.28 |

[a] Length of longest flow path (through sewers) to catchment outfall;

[b] Catchment slope=Difference in ground elevation between upstream most point and outlet / catchment length. This simplistic indictor is used to estimate of whether the catchment exhibits strong slopes on average (Ochoa-Rodriguez et al. 2015). Other types of studies such as ones of surface runoff would indeed require more refined analysis of the topography but they are outside the scope of this paper and refined digital elevation models was not available for all studied areas.

[c] Predominant land use types: R = residential; C = commercial

e The definition (1) is not straightforward due to the loopedness of the catchment

**Table 1: General characteristics of the pilot urban catchments and their semi-distributed urban drainage models**

|  | Sewer system | | | Distributed imperviousness | | $\%_{diff}$[1] | UM parameters for imperviousness map for semi-distributed models | |
|---|---|---|---|---|---|---|---|---|
|  | Outer scale (m) | $D_F$ for large scales | $D_F$ for small scales | Scale of the break | $D_F$ for all scales | % of impervious pixels |  | $\alpha$ | $C_1$ |
| Rotterdam-Centrum | 1024 | 1.94 | 1.07 | 32 | 1.93 | 61 | -9 | 1.29 | 0.017 |
| Rotterdam-Kralingen | 2048 | 1.94 | 1.17 | 64 | 1.89 | 46 | -3 | 0.71 | 0.064 |
| Cranbrook North | 2048 | 1.94 | 0.97 | 64 | 1.83 | 29 | 14 | 1.36 | 0.018 |
| Cranbrook South | 2048 | 1.90 | 0.97 | 64 | 1.81 | 26 | 17 | 1.25 | 0.025 |
| Coimbra West | 512 | 1.90 | 1.03 | 32 | 1.96 | 75 | -18 | 1.37 | 0.009 |
| Ghent North | 2048 | 1.86 | 1.06 | 64 | 1.80 | 24 | 14 | 1.10 | 0.057 |
| Ghent South | 2048 | 1.85 | 1.06 | 64 | 1.82 | 27 | 16 | 1.01 | 0.054 |
| Herent West | 1024 | 1.82 | 1.06 | 64 | 1.71 | 19 | -1 | 1.28 | 0.074 |
| Herent East | 2048 | 1.81 | 1.08 | 64 | 1.72 | 16 | 16 | 0.87 | 0.083 |
| Sucy-en-Brie | 1024 | 1.80 | 1.00 | 64 | 1.79 | 26 | 11 | 1.60 | 0.013 |
| Coimbra East | 512 | 1.79 | 0.97 | 32 | 1.86 | 45 | 13 | 1.71 | 0.20 |
| Jouy-en-Josas | 1024 | 1.79 | 1.79 | 64 | 1.75 | 22 | x | x | x |
| Torquay South | 1024 | 1.77 | 1.77 | 64 | 1.86 | 38 | -16 | 1.45 | 0.062 |

| Torquay North | 1024 | 1.71 | 1.71 | 64 | 1.82 | 29 | -6 | 1.44 | 0.084 |
|---|---|---|---|---|---|---|---|---|---|
| Morée-Sausset | 4096 | 1.69 | 1.69 | 128 | 1.88 | 34 | -1 | 1.64 | 0.023 |

[1] see explanations in last paragraph of section 4.2 and Fig. 9

**Table 2: Estimated fractal dimensions of the sewer system and impervious areas for all the studied areas.**

Figures:

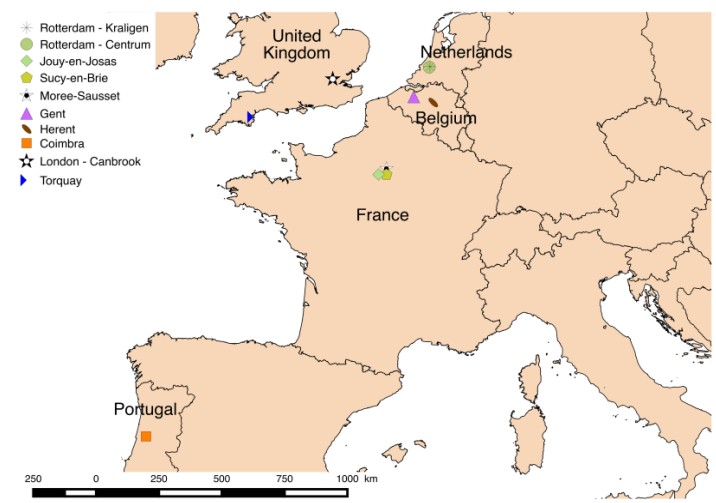

**Figure 1: Location of the pilot urban catchments**

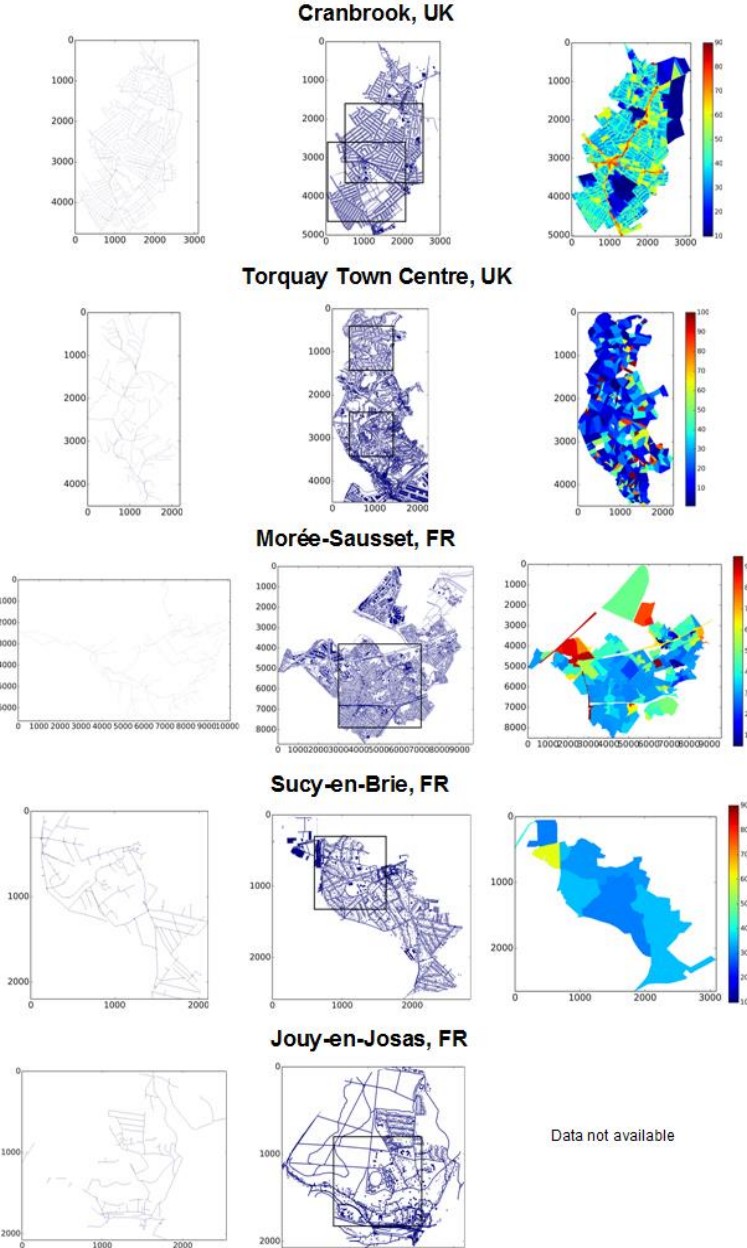

Cranbrook, UK

Torquay Town Centre, UK

Morée-Sausset, FR

Sucy-en-Brie, FR

Jouy-en-Josas, FR

Data not available

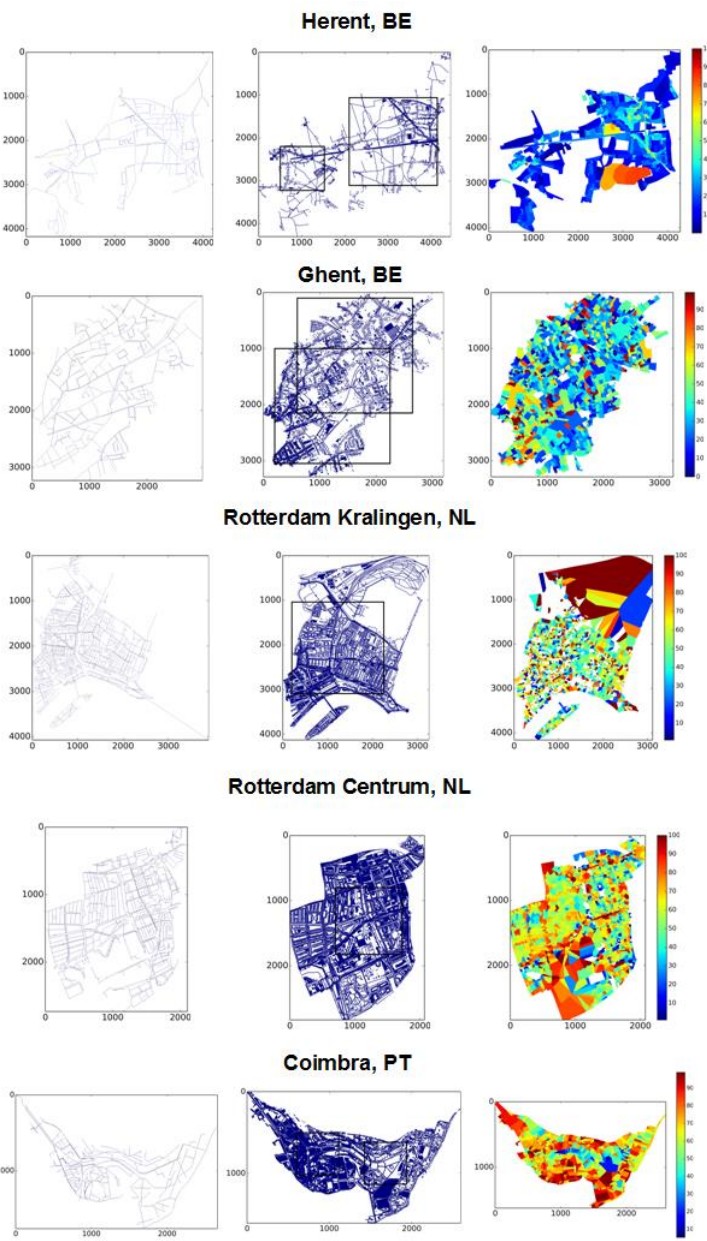

**Figure 2 : Sewer system (left), distributed imperviousness map with pixels of size 2 m (middle) and maps of the imperviousness (%) as assigned to each sub-catchment in the semi-distributed models (right) of the pilot catchments. The axes correspond to meters (m). The black squares (visible in the middle column) correspond to the studied areas in the fractal analysis.**

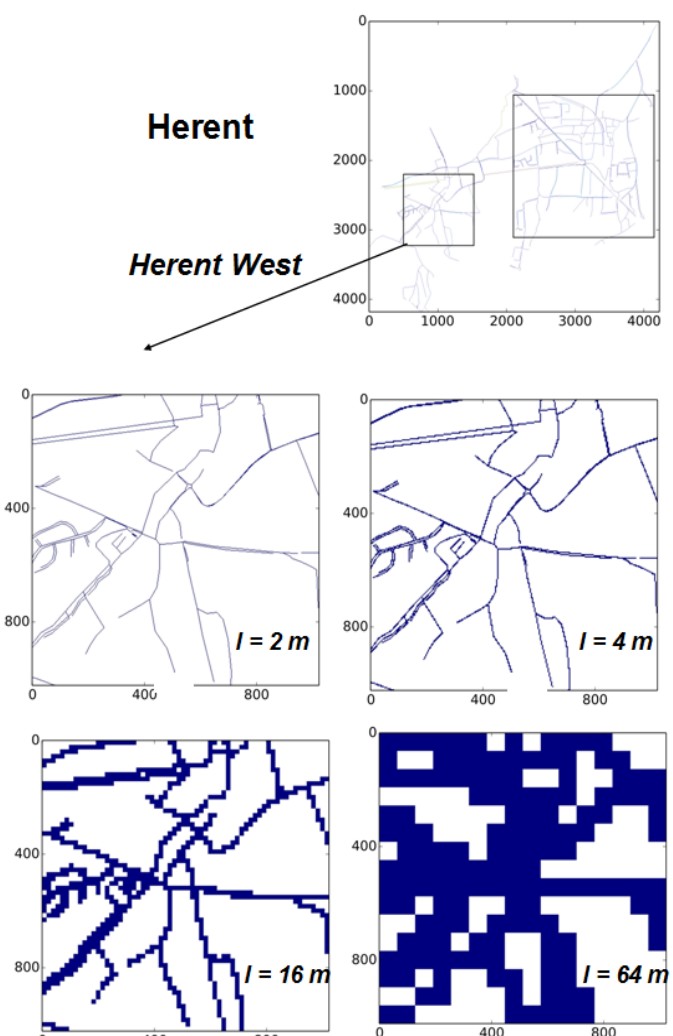

**Figure 3: The sewer network of the Herent West study area observed with the help of pixels of various sizes. The axes correspond to meters (m)**

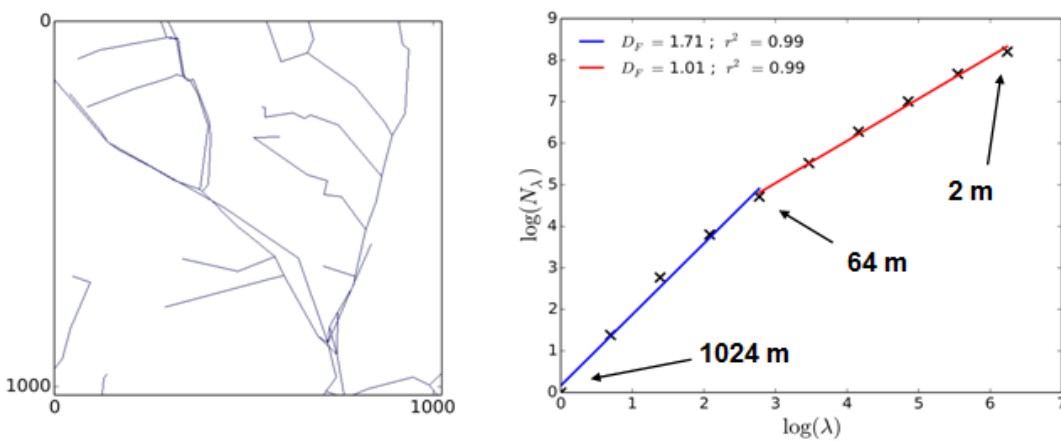

Figure 4: Sewer system (left) and computation of the corresponding fractal dimension, i.e. Eq. 1 in log-log plot (right), for the Torquay North study area. For the left figure, the axes correspond to meters (m).

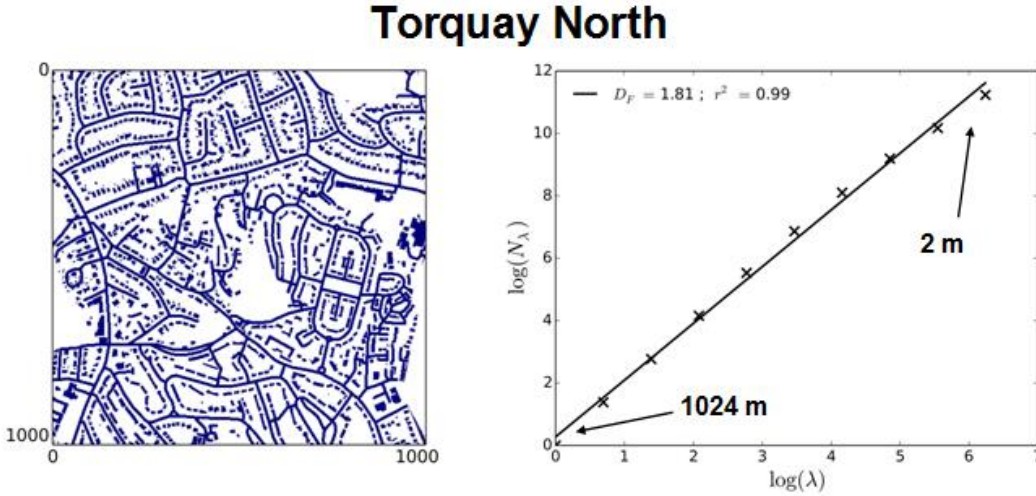

Figure 5: Impervious pixels at a 2 m resolution (left) and computation of the fractal dimension of the corresponding geometrical set, i.e. Eq. 1 in log-log plot, (right) for the Torquay North study area. For the left figure, the axes correspond to meters (m).

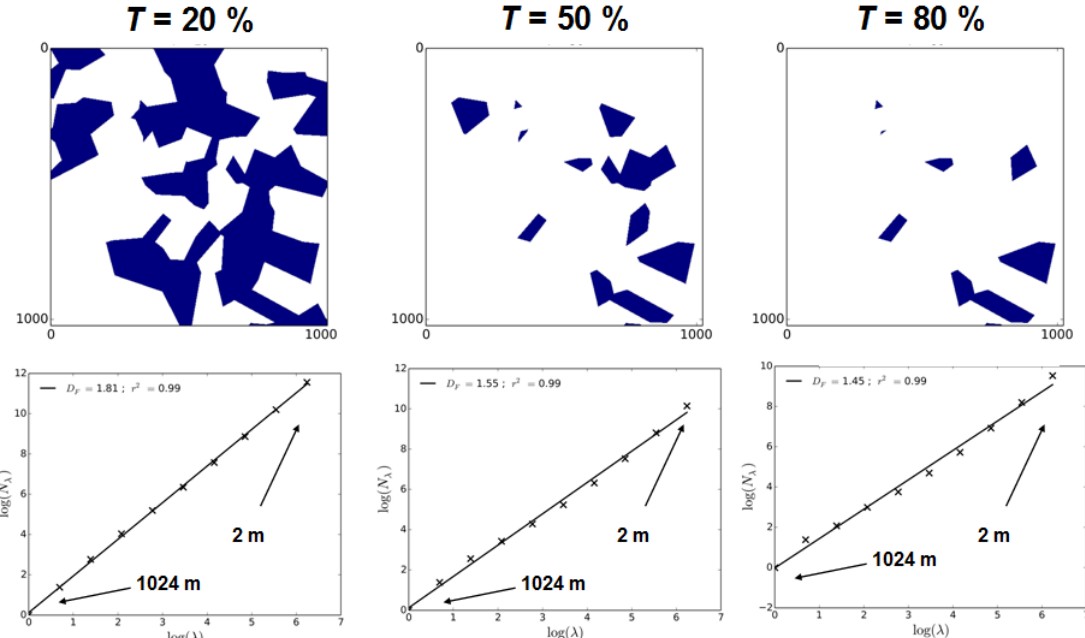

**Figure 6: Illustration of the computation of the fractal dimension of the area covered by the sub-catchments whose imperviousness is greater than a threshold *T* for *T* equal to 20% (left), 50% (middle) and 80% (right) for the Rotterdam-Centrum study area: corresponding geometrical set (top) and Eq. 1 in log-log plot (bottom). For the upper figures, the axes correspond to meters (m)**

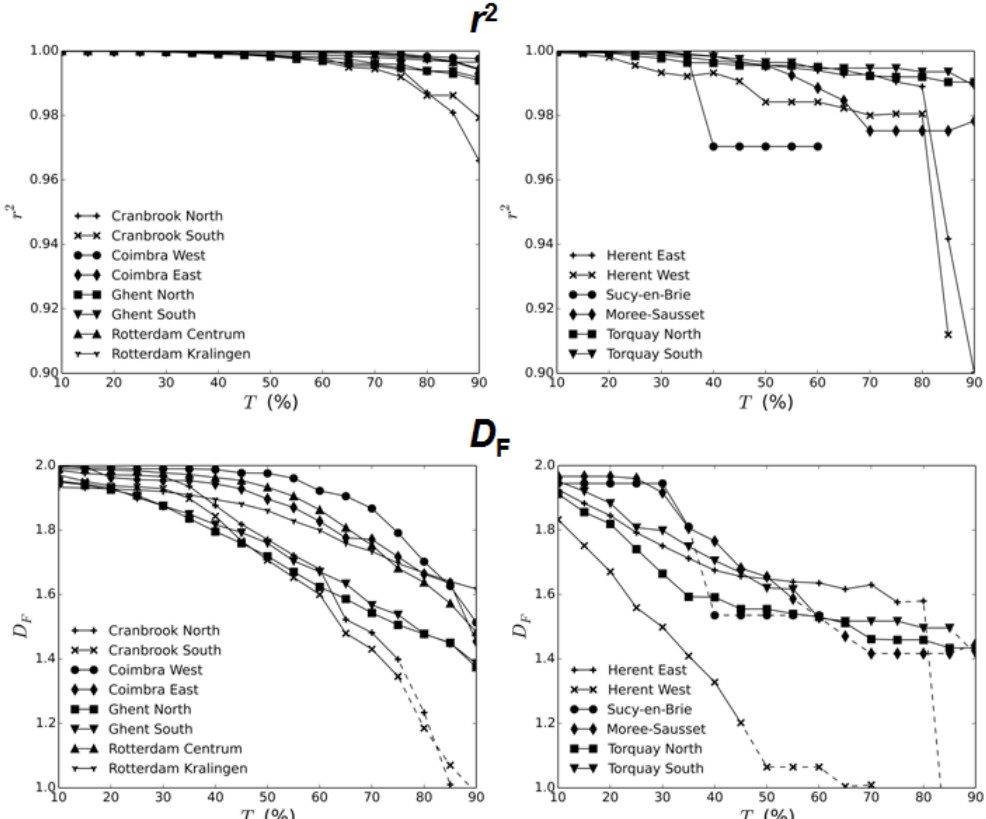

**Figure 7: Fractal dimension analysis of the area covered by the sub-catchments with imperviousness greater than a threshold _T_ for various values of _T_: $r^2$ vs. _T_ (top) and $D_F$ vs. _T_ (bottom). On the bottom curves the dash portions correspond to thresholds for which $r^2 < 0.99$ meaning the estimates are less reliable robust (poorer quality of the scaling). Fractal dimension are computed on the whole range of available scales (i.e. between 2 m and 512 to 4096 m according the study area)**

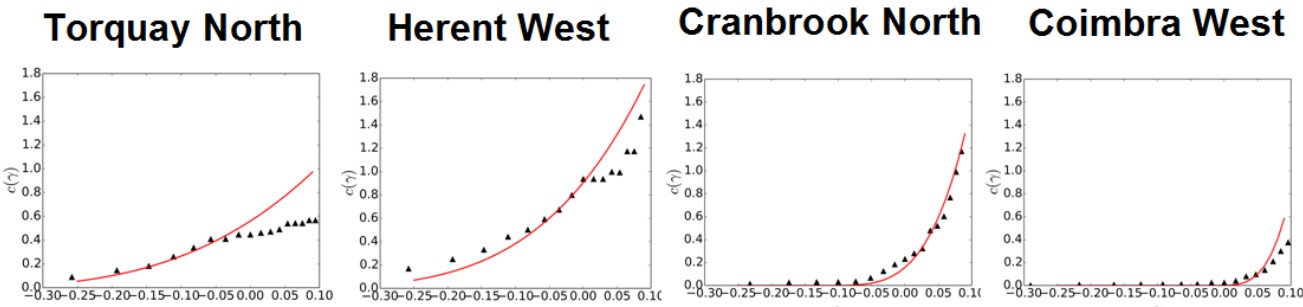

**Figure 8: Functional box counting analysis of the map of sub-catchments imperviousness for 4 selected catchments. Triangles: for each threshold 2-$D_F$ (Fig. 7) vs. the corresponding singularity $\gamma_T$ is estimated as $\log_\Lambda \dfrac{T}{\langle T \rangle}$ (where $\langle T \rangle$ is the average of the**

studied thresholds and equal to 50 here). Solid line: theoretical shape of $c(\gamma)$ with UM parameters estimated with the help of DTM technique (Table 2).

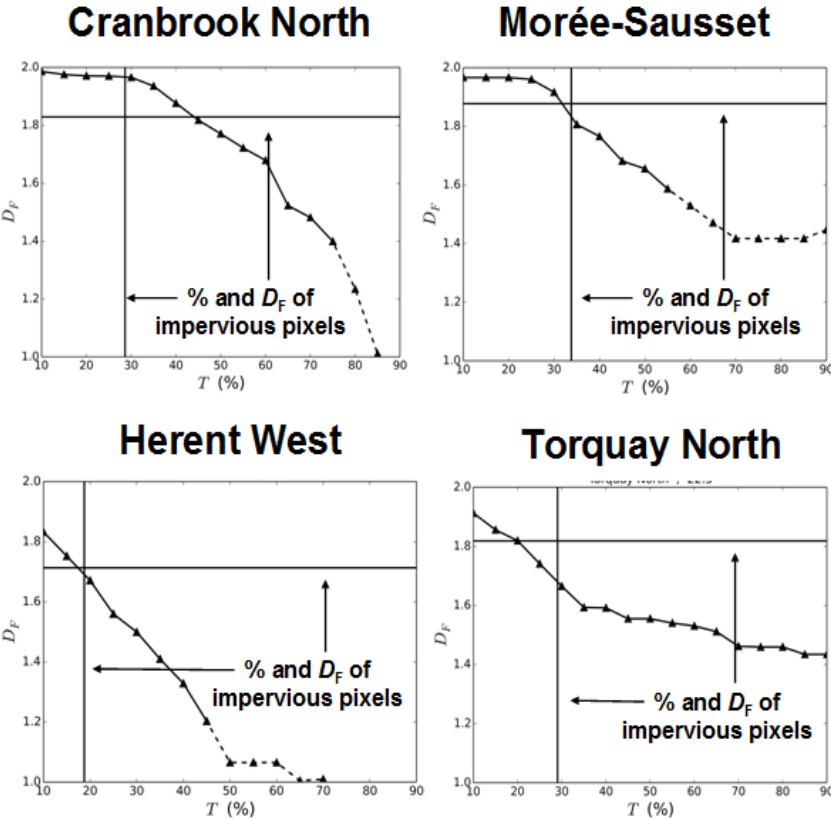

Figure 9: For four study areas: $D_F$ vs. $T$ for the map of sub-catchments imperviousness in model is plotted (same as in Fig. 7), fractal dimension from the distributed data (horizontal line), and percentage of impervious pixel at the two meter resolution (vertical line)

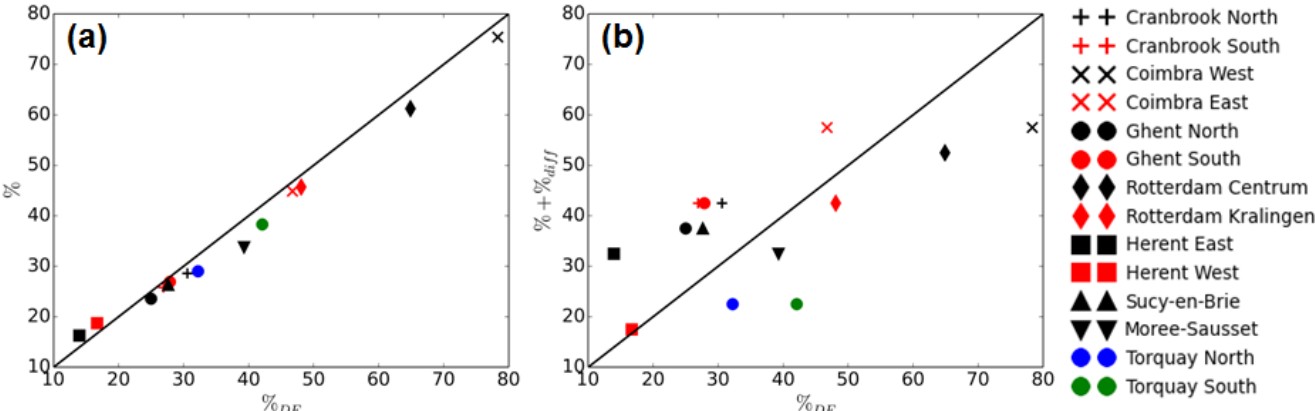

**Figure 10:** The percentages of distributed imperviousness (%) at the highest data resolution (a) and of the imperviousness of semi-distributed models (%+%$_{diff}$) (b) as function of the percentages of imperviousness resulting from the fractal dimension estimates ( %$_{D_F}$ ). The black line indicates the first bisector.

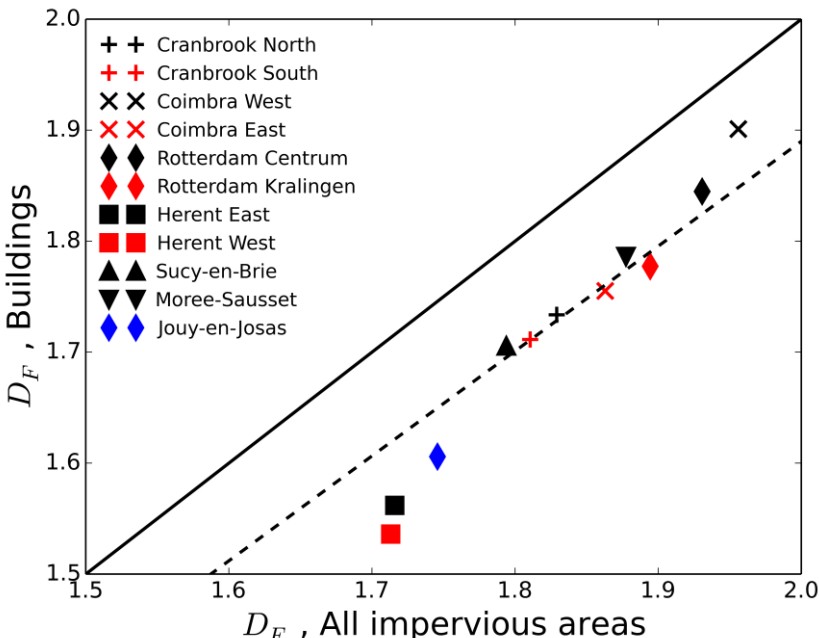

**Figure 11:** Empirical relation between the fractal dimensions of the total impervious area and of buildings only. The continuous line indicates the first bisector, while dotted line is given by: $D_{F\_build} = 0.945 D_{F\_all}$