# Peer review of "Fractal analysis of urban catchments and their representation in semi-distributed models: imperviousness and sewer system"

_Hydrology and Earth System Sciences, 2016_

## Referee Comment (RC1) · S. De Bartolo (Referee) · 18 Nov 2016

The article is well written and provides a very comprehensive fractal analysis framework of urban cathments and their representation in semi-distributed models.

The idea to use a simple scaling analysis in this hydrological field is very interesting, also for the use of a fixed size algorithm procedure to find the fractal dimension as a geometric descriptor. In any case, I invite, the authors also to provide some additional items about the choice of a simple scaling analysis respect to a multifractal (or multi-scaling) analysis . Many recent studies show a multifractal behavior of river networks at large and small scales. Authors can motivate this choice? I suppose that, in the framework of this investigation, the data set type used is very important for the defin-

ion, for example, of the fractal dimension of buildings and also for other sub-cathcment partitions of the hydrological measures. Overall, I think that the manuscript is worthy of being published in Hess.

---

## Referee Comment (RC2) · Anonymous Referee #2 · 25 Jan 2017

General comments:

The topic of this manuscript is potentially interesting for publication in HESS. Yet, I identify a number of weakness in the manuscript, at different levels, as explained below. The whole manuscript should be carefully revised.

1. Spatial patterns analysis using non-linear or fractal analysis is getting more attention in a wide range of disciplines. This manuscript applies this approach to characterizing patterns relevant for urban pluvial drainage. The idea behind this manuscript can be seen as an interesting contribution to the hydrological modelling of rainfall-runoff processes in urban basins, and therefore to the hydraulic design and operation of sewer systems. I encourage the idea of using scaling, (multi)fractal tools to support hydrological studies and appreciate the attempt in this study to conduct this application in urban hydrology.

2. However, I see several weaknesses in this work, and they are of different origin, e.g.: i) technical issues; ii) description of data and models; iii) presentation and interpretation of results; iv) insight into the relation between the mathematical exercise of determining the fractal dimension for a geometrical entity and the physical entity that it represents (in this case vital for the hydrological/engineering application pursued); v) conclusions that are not supported by evidence in the text.

3. In general, the manuscript is written only fairly clearly and there are typos. The overall structure is standard, but there are parts that could be improved. Across the manuscript many parts/paragraphs run in new and new information without a clear main point/focus, which makes the reading difficult and weaknesses the message in the text. The selection of figures, and the figures themselves, should be revised (see below). The literature review could be improved; it misses to include important contributions in the literature related to using scaling frameworks in hydrology.

Specific comments:

4. Data: - The study areas are not adequately described bearing in mind the type of study and application. Key properties of the drainage areas and networks (e.g., sewer density, impervious surfaces, slopes) are also not clearly described, the methods used for its characterization are unclear. - The definition given for the catchment slope (Table 1, footnote 2) suggests that the information in the GIS and available Digital Elevation Models is not fully explored, since it would to be expected that such tools would lead to a much better estimate of catchment slope than the extremely simple one described. - After referring to 10 pilot/study areas, there are figures where some of those areas are unexpectedly split in different parts (e.g. W/E, N/S), and there are situations when simply some of the announced basins are ignored. This should be clarified. - There are basic elements missing: for example, the definition of sub-catchment (SC) is crucial for

the study. - The reference to the basins is not consistent; for example, in the legend of Figure 10, the reference to "Centrum" is inadequate.

5. Methodology: - This section needs to be much improved; there are many confusing sentences. - Proper credit to the methods introduced should be given. For example, a reference for Universal Multifractals is not given (page 5, line 18).

6. Results and discussion: - One technical issue is related to the interpretation of the results obtained with the box-counting method to find the (central, here) fractal dimension. The box-counting plots should be more adequately discussed, including an interpretation for the straight sections in the plot, the cross over between them, the critical scales – the material presented should be carefully checked for misinterpretations. When applying the box-counting method to complex drainage systems (natural or/and artificial drains), there are different issues that need to be better addressed: i) issues related to the resolution of the data set (for example, that only drains above a certain size are displayed) that are known to interfere with the pattern "recognized" by the box-counting algorithm; ii) the space filling character of complex drainage channels that are randomly distributed over the plane and therefore lead to a fractal dimension 2 (all boxes are filled above a critical size; thus, a (box-counting) power-law dependence with a fractal dimension D=2 contains the same level of information as a scale invariant regime with D<2 ?); iii) what would the interpretation be for D in the limit size -> 0? iv) how would the scaling range change with map resolution? Overall, what is the information indeed useful for this urban hydrology application?

- In my view, there is now no clear evidence in the manuscript of the added value of this framework. The authors are not successful in taking the exercise of applying the box-counting method to a geometrical entity - that is highly dependent on the definition of the urban drainage system itself - beyond it: more discussion should be provided that clarify the usefulness of the approach as a practical tool for characterizing (urban, periurban) drainage networks. Examples should provide insight; the ones given are, in my view, not discussed at length. The physical properties of the drainage networks

should be given and this is important for the sake of comparability; the fact that pilot areas/case studies have different origins might imply differences that need to be well established. Although I understand that the authors might have found important to give many different examples (different urban environments, from different countries), for full understanding of the results the physical interpretation should be allowed – and this implies a good definition of the systems. Per se, a list of fractal dimensions estimated for not well understood drainage systems is of very limited interest to the community.

- It also does not help the manuscript the selection of figures included therein. The selection of figures should be better though. The present selection of figures does not allow the reader to follow the analysis and characterization for one of the study areas. Every time (i.e. for different figure's content, methods) different cases (study areas) are show, in an apparently ad hoc mix, which does not allow one to attempt reasoning about one single drainage system, and therefore to interpret results.

- For comparison purposes, there is clearly a mismatch between having aprox. 2000 SC or 9 SC in different study areas; this should have been carefully discussed because there is no evidence in the manuscript on the origin of such huge difference that can be thought as resulting from differences in the geomorphology of the basins or simply from the definition of the study unit (SC). The practical implications need more attention.

-There is no attempt to validate if the assessment of imperviousness proposed in this work is more successful than the approach routinely used in the model.

-It should be clarified if the same hydrodynamic model was used for all study areas, or if there are different models being applied. In fact, such model(s) is(are) not described. The type of model would highlight the relative importance of the input, such as the description of the fractal dimension of the drainage network or the fractal dimension of the impervious soil cover spatial pattern. It is not discussed at what resolution should the hydrodynamic model (or models) run for adequate output. The evaluation of the model performance is also not carried out or discussed.

-For the hydrologic/engineering application in mind, qualitative assessments (e.g., page 4, line 5-6, "slight difference" or "more pronounced") should be converted to quantitative assessments.

- There is (apparently) too much reliance on the magnitude of the regression coefficient (r2) and not enough attempt to explain the "meaning" of some results (e.g., page 10, line 4).

-The reference (last paragraph in section 4) to green roof tops is marginal to this work, this reference is not well embedded in the text.

8. Conclusions: - This section includes statements that lack support from results and discussion, as they are presented in the manuscript. - Despite the attempt to include this work in a framework of applied hydrology, in my opinion the work is not successful in providing insight into the application of the scaling analysis proposed in the context of the physical characteristics of an urban pluvial sewer system, which are key in engineering applications.

9. References: - There is mismatch between the references' list and the works cited in the text. - Many references in the list are not complete, there is missing information about the publication (e.g. journal name). - When several references are given in the text, they are not organized alphabetically for chronologically.

Technical corrections

(not exhaustive list) - Page 2, line 18. Should be: Lovejoy and Mandelbrot, 1985. - Page 2, line 28: the reference to fractal analysis of soil features is not relevant for this study. - Page 4, line 5: revise ". . .the values of imperviousness are uniform. . ."; also, Fig. 2 shows "proportion of imperviousness" - Page 4, line 15-16: - Table 1: Numbers used to call footnotes can be confused with powers – change notation. Within each column, there should be consistency in the way the values are presented. - Tables 1 and 2: The two tables organization should match, to allow easy cross reading. -

Figure 2: the distortion introduced by the different scales used in the 2-directional axes is confusing, for the sake of comparability between the different panels in this figure. Also, the reference in the caption that "The axes correspond to the number of 2 m pixels" might be confusing: does this mean that the distance along one axis between 0 and 100 corresponds to 200 m? One expects to have distances along the axes. - Figures 3 to 6: Units for the axes are missing. - Figure 6: in the caption, The reference is to Eq. 1. As one expects distances to be represented in the upper panels, it is unexpected to have the bottom panels referring to distances (1024 m) that are larger than the (origin) data represented above (?). The same occurs in other figures. Maps should provide distances. - Captions of the figures need to provide understanding of the material plotted. For example, the fractal dimension in Fig. 7 was estimated for which scaling range?

———————————————————

---

## Author Comment (AC1) · 8 Mar 2017

S. De Bartolo (Referee)
samuele.debartolo@unical.it

The article is well written and provides a very comprehensive fractal analysis framework of urban cathments and their representation in semi-distributed models. The idea to use a simple scaling analysis in this hydrological field is very interesting, also for the use of a fixed size algorithm procedure to find the fractal dimension as a geometric descriptor. In any case, I invite, the authors also to provide some additional items about the choice of a simple scaling analysis respect to a multifractal (or multi-scaling) analysis. Many recent studies show a multifractal behavior of river networks at large and small scales. Authors can motivate this choice? I suppose that, in the framework of this investigation, the data set type used is very important for the definion, for example, of the fractal dimension of buildings and also for other sub-cathcment partitions of the hydrological measures. Overall, I think that the manuscript is worthy of being published in Hess."

We chose to rely on the use of simple scaling for the analysis of river networks because it enabled to use the same formalism on both river networks and also maps of distributed imperviousness. The reviewer is correct that multifractals have been used to characterize river networks and additional references are now discussed in the introduction. We also would like to stress that multifractals (computation of Universal Multifractals parameters, co-dimension function) were used in the analysis of the representation of imperviousness in semi-distributed models.

---

## Author Comment (AC2) · 8 Mar 2017

General comments:
The topic of this manuscript is potentially interesting for publication in HESS. Yet, I identify number of weakness in the manuscript, at different levels, as explained below. The whole manuscript should be carefully revised.
1. Spatial patterns analysis using non-linear or fractal analysis is getting more attention in a wide range of disciplines. This manuscript applies this approach to characterizing patterns relevant for urban pluvial drainage. The idea behind this manuscript can be seen as an interesting contribution to the hydrological modelling of rainfall-runoff processes in urban basins, and therefore to the hydraulic design and operation of sewer systems. I encourage the idea of using scaling, (multi)fractal tools to support hydrolog-ical studies and appreciate the attempt in this study to conduct this application in urban hydrology.
2. However, I see several weaknesses in this work, and they are of different origin, e.g.: i) technical issues; ii) description of data and models; iii) presentation and interpretation of results; iv) insight into the relation between the mathematical exercise of determining the fractal dimension for a geometrical entity and the physical entity that it represents (in this case vital for the hydrological/engineering application pursued); v) conclusions that are not supported by evidence in the text.
3. In general, the manuscript is written only fairly clearly and there are typos. The overall structure is standard, but there are parts that could be improved. Across the manuscript many parts/paragraphs run in new and new information without a clear main point/focus, which makes the reading difficult and weaknesses the message in the text. The selection of figures, and the figures themselves, should be revised (see below). The literature review could be improved; it misses to include important contributions in the literature related to using scaling frameworks in hydrology.
Specific comments:
4. Data:
- The study areas are not adequately described bearing in mind the type of study and application. Key properties of the drainage areas and networks (e.g., sewer density, impervious surfaces, slopes) are also not clearly described, the methods used for its characterization are unclear."

Following the reviewer remark, we chose to be more exhaustive and to include a figure of all three available data sets for the 10 studied urban areas. Hence Fig. 2 was completed. This should give the reader the additional insight needed.

"- The definition given for the catchment slope (Table 1, footnote 2) suggests that the information in the GIS and available Digital Elevation Models is not fully explored, since it would to be expected that such tools would lead to a much better estimate of catchment slope than the extremely simple one described."

The reviewer is correct that the catchment slope is an extremely simple indicator. It is indeed only an average indicator at the scale of the basin that was designed for a previous analysis

involving the same catchments with other goals (Ochoa-Rodriguez et al. 2015). It was re-used here as an indicator of whether the catchment exhibits strong slopes on average. Other types of studies such as ones of surface runoff would indeed require more refined analyses of the topography but they are outside the scope of this paper. This point was clarified in the caption of Table 1. Furthermore refined DEM was not available for all the catchments.

"- After referring to 10 pilot/study areas, there are figures where some of those areas are unexpectedly split in different parts (e.g. W/E, N/S), and there are situations when simply some of the announced basins are ignored. This should be clarified."
This is actually a technical reason associated with the fractal analysis. In this implementation mode, the analysis requires the use of square areas whose size should be a power of 2. This was actually explained in section 3 on methodology. The terminology "study area" was adopted to avoid confusion. The analysis of imperviousness representation in operational models was not done for the Jouy-en-Josas case because no model was available. This was stated in section 2.

"- There are basic elements missing: for example, the definition of sub-catchment (SC) is crucial for the study."
The definition given in section 2 was completed following the reviewers remark.

"- The reference to the basins is not consistent; for example, in the legend of Figure 10, the reference to "Centrum" is inadequate."
Thank you for your careful reading! This was checked, and captions of figures 7, 10 and 11 were updated.

"5. Methodology: - This section needs to be much improved; there are many confusing sentences. - Proper credit to the methods introduced should be given. For example, a reference for Universal Multifractals is not given (page 5, line 18)."
As suggested by the reviewer the whole section was improved and additional references were added for each of the methods introduced.

"6. Results and discussion:
- One technical issue is related to the interpretation of the results obtained with the box-counting method to find the (central, here) fractal dimension. The box-counting plots should be more adequately discussed, including an interpretation for the straight sections in the plot, the cross over between them, the critical scales"
That is actually what we had tried to do before but this is now improved in section 4.1 and 4.2. An interpretation for each scaling regime is now systematically provided. As an example, for the sewer system small scales only reflect the linear structure of the pipes while large scales regime reflects the space filled by the network.

"– the material presented should be carefully checked for misinterpretations. When applying the box-counting method to complex drainage systems (natural or/and artificial drains), there are different issues that need to be better addressed: i) issues related to the resolution of the data set (for example, that only drains above a certain size are displayed) that are known to interfere with the pattern "recognized" by the box-counting algorithm; ii) the space filling character of complex drainage channels that are randomly distributed over the plane and therefore lead to a fractal dimension 2 (all boxes are filled above a critical size; thus, a (box-counting) power-law dependence with a fractal dimension D=2 contains the same level of information as a scale invariant regime with D<2 ?); iii) what would the interpretation be for

D in the limit size -> 0? iv) how would the scaling range change with map resolution? Overall, what is the information indeed useful for this urban hydrology application?"

Thanks to the reviewer suggestions, these important issues are now discussed in the new version of the manuscript (section 4.1). i) Indeed the reviewer is correct. This point is visible in this analysis through the location of the scaling break in the sewer analysis. Considering only larger pipes will lead to shifting this scale break to larger scales. ii) It was clarified that scaling regime with Df <2 are found for the data sets studied. iii) For pixel size → 0, in the case of the sewer system it would simply extend the width of the range of scales for the small scales regime with Df=1. In the case of distributed imperviousness one can expect the unique scaling regime to continue down to the physical size of the structures below which Df=2 would be found. iv) Changing the initial resolution of the map would extend or shrink the width of the small scales regime for the sewer system and not affect the estimates of Df for the distributed imperviousness (see limitation in iii)) since a unique scaling regime is retrieved. The added value to urban hydrology by the definition of robust indicator of the level of urbanization was clarified.

"- In my view, there is now no clear evidence in the manuscript of the added value of this framework. The authors are not successful in taking the exercise of applying the box-counting method to a geometrical entity - that is highly dependent on the definition of the urban drainage system itself - beyond it: more discussion should be provided that clarify the usefulness of the approach as a practical tool for characterizing (urban, periurban) drainage networks. Examples should provide insight; the ones given are, in my view, not discussed at length. The physical properties of the drainage networks should be given and this is important for the sake of comparability; the fact that pilot areas/case studies have different origins might imply differences that need to be well established. Although I understand that the authors might have found important to give many different examples (different urban environments, from different countries), for full understanding of the results the physical interpretation should be allowed – and this implies a good definition of the systems. Per se, a list of fractal dimensions estimated for not well understood drainage systems is of very limited interest to the community."
Showing that fractal dimension can be used to characterize the drainage network of an area is indeed one of the results of the paper. But another one (and more innovative) is that the fractal dimensions found on sewer network and distributed imperviousness are similar for all the studied areas. We believe that the confirmation of this similarity on "many different examples" (as mentioned by the reviewer) actually reinforce it. This point was clarified in section 4.1 and the conclusion.

"- It also does not help the manuscript the selection of figures included therein. The selection of figures should be better though. The present selection of figures does not allow the reader to follow the analysis and characterization for one of the study areas. Every time (i.e. for different figure's content, methods) different cases (study areas) are show, in an apparently ad hoc mix, which does not allow one to attempt reasoning about one single drainage system, and therefore to interpret results."
The initial idea was to display a graph for each of the study areas. However following the interesting reviewer comment, the strategy was completely changed and figures re-plotted. In the revised version, all the curves are plotted for one study area (Torquay North). Additional ones for other areas are provided when needed for the discussion.

"- For comparison purposes, there is clearly a mismatch between having aprox. 2000 SC or 9 SC in different study areas; this should have been carefully discussed because there is no evidence in the manuscript on the origin of such huge difference that can be thought as resulting from differences in the geomorphology of the basins or simply from the definition of the study unit (SC). The practical implications need more attention."

The choice of this paper was to analyse the inputs of models that are actually used operationally by the people managing urban drainage in these areas (Ochoa-Rodriguez et al. 2015). They have various approaches leading indeed to different choices of typical sub-catchment sizes. This was clarified in the data presentation. As pointed by the reviewer this has practical implications on the analysis carried out, basically limiting the possible interpretation of the curve Df vs. T. This point is now reinforced in the new version (section 4.2). A discussion on the practical implications, i.e. making a case for higher resolution modelling was also added in section 4.2.

"-There is no attempt to validate if the assessment of imperviousness proposed in this work is more successful than the approach routinely used in the model."

Contrary to other formalisms such as the use of a single percentage of imperviousness defined with data at an arbitrary scale, this fractal dimension is a quantity valid across scales and furthermore based on the characterization of two aspects related to urbanization (namely the sewer network and the distributed imperviousness) which makes it robust. This was clarified in the discussion of section 4.1.

"-It should be clarified if the same hydrodynamic model was used for all study areas, or if there are different models being applied. In fact, such model(s) is(are) not described. The type of model would highlight the relative importance of the input, such as the description of the fractal dimension of the drainage network or the fractal dimension of the impervious soil cover spatial pattern. It is not discussed at what resolution should the hydrodynamic model (or models) run for adequate output. The evaluation of the model performance is also not carried out or discussed."

The models are not the same for all the pilot sites but they all function with the same underlying principles. The description of the functioning of these semi-distributed models was refined in section 2. The evaluation of model performance is outside the scope of this paper which focuses on the use of fractal tools to characterize urban environment with an extension to the representation of imperviousness of operational models. The model "validation" was carried out previously by the practitioners using them. Interested reader can found results with simulation outputs in Ochoa-Rodriguez et al. 2015. This was clarified in the data presentation section.

"- For the hydrologic/engineering application in mind, qualitative assessments (e.g., page 4, line 5-6, "slight difference" or "more pronounced") should be converted to quantitative assessments."

The reviewer in correct that qualitative assessments should be avoided. With regards to the example mentioned (p.5 l.5-6), it was clarified that the comments were made based on visual inspection of the figures and furthermore added that they are actually consistent with the scale break at 64 m that is identified and discussed in section 4.1.

"- There is (apparently) too much reliance on the magnitude of the regression coefficient (r2) and not enough attempt to explain the "meaning" of some results (e.g., page 10, line 4)."

Indeed in this example the differences of r2 are not great enough to enable an interpretation. Hence the sentence was rephrased to simply say that they confirm the quality of the scaling behaviour.

"- The reference (last paragraph in section 4) to green roof tops is marginal to this work, this reference is not well embedded in the text."
This discussion was added because green roofs are one of the main tool to act on urban flow if needed. This is now clearly stated in the mentioned paragraph (section 4.3).

"8. Conclusions: - This section includes statements that lack support from results and discussion, as they are presented in the manuscript. - Despite the attempt to include this work in a framework of applied hydrology, in my opinion the work is not successful in providing insight into the application of the scaling analysis proposed in the context of the physical characteristics of an urban pluvial sewer system, which are key in engineering applications."

Following the reviewer's comment, the conclusion was changed and now clearly distinguishes the findings supported by the results of this paper and the perspectives for future work. In the initial version, this distinction was indeed not sufficiently straightforward and could lead to misinterpretations.

"9. References: - There is mismatch between the references' list and the works cited in the text. - Many references in the list are not complete, there is missing information about the publication (e.g. journal name). - When several references are given in the text, they are not organized alphabetically for chronologically."
This was checked and updated.

"Technical corrections
(not exhaustive list) - Page 2, line 18. Should be: Lovejoy and Mandelbrot, 1985."
This was done

"-Page 2, line 28: the reference to fractal analysis of soil features is not relevant for this study."
We included it to show the variety of use of the fractal dimension in the field of hydrology.

"- Page 4, line 5: revise ": : :the values of imperviousness are uniform: : :"; also, Fig. 2 shows "proportion of imperviousness""
This was done and caption of Fig. 2 updated.

"- Page 4, line 15-16: - Table 1: Numbers used to call footnotes can be confused with powers – change notation. Within each column, there should be consistency in the way the values are presented."
The notation for footnotes was changed to letters to avoid confusion. We do not really understand your point on consistency. The case of the computation of "Length" for Rotterdam is explained in a footnote.

"- Tables 1 and 2: The two tables organization should match, to allow easy cross reading."
Actually Table 1 describes pilot sites and table 2 study areas (see explanation in a previous answer). Table 2 is sorted with decreasing order of the fractal dimension of sewer network.

- Figure 2: the distortion introduced by the different scales used in the 2-directional axes

is confusing, for the sake of comparability between the different panels in this figure. Also, the reference in the caption that "The axes correspond to the number of 2 m pixels" might be confusing: does this mean that the distance along one axis between 0 and 100 corresponds to 200 m? One expects to have distances along the axes. - Figures 3 to 6: Units for the axes are missing. - Figure 6: in the caption, The reference is to Eq. 1. As one expects distances to be represented in the upper panels, it is unexpected to have the bottom panels referring to distances (1024 m) that are larger than the (origin) data represented above (?). The same occurs in other figures. Maps should provide distances.

All the figures were updated and re-plotted to include axes expressed in m to avoid the confusion pointed out by the reviewer.

- Captions of the figures need to provide understanding of the material plotted. For example, the fractal dimension in Fig. 7 was estimated for which scaling range?

It was clarified that fractal dimension are computed on the whole range of available scales (i.e. between 2 m and 512 to 4096 m according the study area). The other captions were checked.

---

## Editor Decision (ED1)

**Final corrections**

| Page | Line | Correction |
|---|---|---|
| 3 | 11 | the bracket should not be before 'De Bartolo' but before '2004' |
| 4 | 21 | 'Rodriguez' instead of 'Ridriguez' |
| 5 | 26 | 'such as a' instead of 'such a' |
| 6 | 2 | 'where' instead of 'Where' |
| 6 | 24 | add 'of' before 'the scale range' |
| 6 | 27 | 'Morée' instead of 'Moree' |
| 9 | 20 | add 'it' before 'was not possible' |
| 11 | 9 | 'compared' instead of 'compare' |
| 11 | 12 | 'roofs' instead of 'roof' |
| 11 | 28 | 'of identifying' instead of 'to indentify' |
| 12 | 12 | 'perspectives' instead of 'perspective' |